# HiPro-CT: A Hierarchical Probabilistic Framework for 3D Medical Vision-Language Alignment

**Lin Lu**[1]        L-LU22@MAILS.TSINGHUA.EDU.CN
**Zihan Liu**[1]        ZH-LIU22@MAILS.TSINGHUA.EDU.CN
**Chaoxiang Tang**[1]        TANG-CX24@MAILS.TSINGHUA.EDU.CN
**Hui Zhang**[*1]        HZHANG@TSINGHUA.EDU.CN

[1] *School of Biomedical Engineering, Tsinghua University, Beijing, China*

**Editors:** Accepted for publication at MIDL 2026

## Abstract

The adaptation of vision-language models (VLMs) to 3D medical imaging is currently impeded by two fundamental bottlenecks: the dilution of local features caused by the granularity mismatch between volumetric data and textual reports, and the inability of deterministic embeddings to capture the inherent semantic uncertainty of clinical descriptions. To address these challenges, we propose HiPro-CT, a novel hierarchical probabilistic framework for 3D medical vision-language alignment. Unlike traditional point-based approaches, HiPro-CT maps images and texts into Gaussian probability distributions, utilizing variance to explicitly quantify uncertainty and enhance robustness against incompleteness and polysemy. We introduce a soft masked pooling strategy that performs weighted feature aggregation guided by anatomical masks, enabling precise organ-level alignment while preserving boundary context. Furthermore, we devise a hierarchical inclusion loss to enforce geometric constraints within the embedding space, ensuring that the deterministic global representations are geometrically grounded within the strictly more uncertain local distributions. Extensive experiments demonstrate that HiPro-CT significantly outperforms state-of-the-art deterministic baselines in zero-shot multi-abnormality detection and cross-modal retrieval, validating the efficacy of integrating fine-grained anatomical supervision with probabilistic representation learning.

**Keywords:** Vision-Language Models, Computed Tomography, Probabilistic Embedding, Uncertainty Modeling, Fine-grained Alignment

## 1. Introduction

In recent years, the application of deep learning in medical image analysis has evolved from single-modality, single-task approaches toward multi-modal, general-purpose Foundation Models. Vision-Language Models (VLMs), represented by CLIP (Contrastive Language-Image Pre-training) (Radford et al., 2021), have demonstrated remarkable open-vocabulary recognition capabilities through pre-training on large-scale image-text pairs. However, transferring this paradigm to three-dimensional medical imaging, such as CT and MRI, encounters two fundamental challenges.

First, the mismatch in information granularity between vision and text leads to feature dilution. There is a substantial disparity in information density between medical images and radiology reports. A single 3D CT volume contains millions of voxels replete with rich

---

* Corresponding author

anatomical details, whereas the corresponding report typically consists of only hundreds of words, highly focused on pathological findings. Existing mainstream models (e.g., MedCLIP (Wang et al., 2022), CT-CLIP (Hamamci et al., 2024)) usually adopt a coarse-grained global alignment strategy, compressing high-dimensional volumetric data into a single vector via global pooling. This operation inevitably leads to feature dilution, causing critical local features to be overshadowed by background anatomical structures, thereby hindering precise semantic alignment.

Second, medical text is characterized by semantic incompleteness and uncertainty. Medical image-text pairs possess inherent "one-to-many" polysemy (for instance, the same imaging manifestation may be described as a "nodule" or a "mass", while reports from different patients may contain identical sentences). Traditional deterministic embeddings attempt to map complex clinical semantics to a single point in Euclidean space. Mathematically, this often leads to embedding space collapse, failing to effectively capture the inherent range of uncertainty in clinical diagnosis. Furthermore, such deterministic mapping struggles to handle the "false negative" supervision signals arising from the omission of normal organ descriptions in reports, thus limiting the model's ability to learn representations of normal anatomical structures.

To alleviate these challenges, this study proposes **HiPro-CT**, constructing the first 3D medical VLM that combines fine-grained strong supervision with probabilistic modeling. The core philosophy is to introduce probabilistic representations and hierarchical geometric constraints atop fine-grained visual alignment, thereby uniformly characterizing the semantics and uncertainty of both images and text while explicitly injecting anatomical hierarchical information. The specific innovations are as follows:

- **Probabilistic embedding:** Images and texts are elevated from point vectors to Gaussian probability distributions. The distribution mean carries the core semantics, while the variance quantifies uncertainty. This enhances the model's robustness to polysemy and semantic ambiguity and provides a more rational metric for cross-modal similarity.

- **Soft masked pooling:** Instead of hard cropping, organ or regional masks are utilized to perform soft weighting on features. This approach preserves boundary and contextual information while calculating organ-level features, enabling finer-grained semantic alignment.

- **Hierarchical inclusion loss:** Geometric constraints are introduced within the probabilistic space to ensure that local organ distributions, which contain less information, statistically "encompass" the global distribution. This mechanism explicitly encodes the "global-organ" anatomical semantic hierarchy and calibrates representations across different scales.

Subsequently, we will systematically review the relevant literature, focusing on the evolution of medical VLM architectures, cross-modal applications of probabilistic representation learning, fine-grained visual alignment strategies, and hierarchical semantic constraint mechanisms, to substantiate the theoretical basis and innovative value of HiPro-CT.

## 2. Related Work

### 2.1. 3D Medical Vision-Language Models: From Global to Fine-grained

Early medical Vision-Language Models (VLMs) primarily focused on 2D modalities, such as X-rays (Shentu and Al Moubayed, 2024; Mamdouh et al., 2025). Pioneering works like MedCLIP and BiomedCLIP (Zhang et al., 2023) adapted CLIP-style contrastive learning to the medical domain, employing decoupled dual-tower encoders to achieve efficient zero-shot classification and open-vocabulary recognition. To accommodate 3D modalities like CT and MRI, models such as CT-CLIP and Med3DVLM (Xin et al., 2025) introduced 3D Transformers or 3D CNNs to encode entire volumetric data into detailed global features for alignment with radiology reports. However, "Global-Global" alignment reveals significant information asymmetry in three-dimensional scenarios: massive voxel details are compressed via coarse-grained pooling, causing subtle pathological features to be diluted by background noise, which hinders the model's ability to perceive minute anomalies.

To alleviate this bottleneck, researchers have begun exploring fine-grained alignment mechanisms. CT-GLIP (Lin et al., 2024), drawing inspiration from Grounded Language-Image Pre-training, utilizes pre-computed segmentation masks to perform hard cropping. This forces the alignment of specific organ image patches with corresponding descriptive sentences, thereby enhancing organ-level recognition capabilities. Nevertheless, hard cropping relies heavily on high-quality boundaries and tends to discard peripheral context, limiting performance in scenarios involving invasive tumors or blurred boundaries. Anatomy-VLM (Gu et al., 2025) addresses this by extracting regions of interest (ROI) at the feature map level and concatenating local and global tokens to enhance context awareness. However, most such approaches remain based on deterministic bounding boxes or binary masks. They struggle to characterize the "unmentioned $\neq$ non-existent" nature of reports and terminological polysemy, failing to explicitly model uncertainty and anatomical hierarchy at the representation level.

### 2.2. Probabilistic Representation and Uncertainty Modeling

To address semantic ambiguity and the "one-to-many" nature of cross-modal matching, probabilistic embedding has progressively entered vision-language research. PCME (Chun et al., 2021; Chun, 2023) pioneered the embedding of images and texts as probability distributions (typically Gaussian), where the distribution mean represents central semantics and the variance characterizes range and uncertainty. By replacing point-to-point similarity with probabilistic similarity, this approach improves robustness in many-to-many matching. ProLIP (Chun et al., 2024) further introduced an "uncertainty token" for efficient variance estimation and utilized an "inclusion loss" to learn geometric relationships between occluded and complete information, thereby improving cross-modal retrieval and descriptive consistency.

In the medical domain, however, probabilistic modeling remains in its nascent stages. ProbMED (Gao et al., 2025) utilized Hellinger Distance instead of cosine similarity to measure distributional divergence, validating the efficacy of probabilistic methods in multimodal alignment for chest X-rays and electrocardiograms. Anatomy-VLM (Gu et al., 2025), functioning on the premise that "text is more abstract," adopted von Mises-Fisher distri-

butions for post-hoc modeling of text embeddings to mitigate textual uncertainty. Nevertheless, existing medical VLMs often limit probabilistic modeling to a single modality (predominantly text), overlooking visual uncertainty in 3D medical imaging caused by imaging quality, motion artifacts, or blurred lesion boundaries. Furthermore, they lack a unified probabilistic framework synergistic with fine-grained regional alignment and fail to explicitly inject anatomical hierarchical structures within the probabilistic space.

In summary, existing 3D medical VLMs are constrained by two primary bottlenecks. First, fine-grained alignment relies excessively on hard cropping, making it difficult to balance contextual information around lesions with boundary robustness. Second, probabilistic modeling has not yet successfully covered both image and text modalities, nor has it explicitly constrained the "Global-Local" anatomical hierarchy at the distributional level. Addressing these gaps, we propose a unified framework combining Soft Masked Pooling with Hierarchical Probabilistic Embeddings. This approach aims to transcend current performance ceilings in open-vocabulary recognition and small lesion detection through more precise semantic alignment and effective uncertainty characterization.

## 3. Materials and Methods

### 3.1. Datasets

In this study, we utilize the RadGenome-Chest CT dataset (Zhang et al., 2024). Built upon 25,692 non-contrast 3D chest CT volumes and their corresponding reports from the CT-RATE repository, this dataset was originally constructed via a dual-stream pipeline designed to achieve region-level alignment and augmentation between visual and textual modalities.

On the imaging side, the dataset employed a text-prompted universal segmentation model (SAT) (Zhao et al., 2023) to perform comprehensive 3D segmentation across 197 chest-related anatomical categories. The volumetric data were standardized to a unified voxel spacing of $1\,\text{mm} \times 1\,\text{mm} \times 3\,\text{mm}$, resulting in a multi-level anatomical structure tree with corresponding organ-level masks.

On the textual side, a report sentence splitting and classification model was applied to decompose the findings and impression sections into region-specific descriptions organized by anatomical hierarchy. This process established explicit anchoring relationships between individual sentences and their corresponding segmentation masks. Consequently, the dataset provides approximately 665,000 multi-granularity, sentence-level "region-report" pairs for training and evaluation.

### 3.2. Framework

As illustrated in Figure 1, the proposed HiPro-CT is a unified framework designed for hierarchical probabilistic vision-language alignment. The architecture consists of parallel 3D visual and textual encoders that map input CT volumes and radiology reports into a shared probabilistic embedding space, modeling them as Gaussian distributions to capture inherent uncertainty. To bridge the granularity gap, the framework incorporates two distinct alignment levels: a global branch for aligning the entire volume with the full report, and a fine-grained branch utilizing Soft Masked Pooling to align specific organ features with

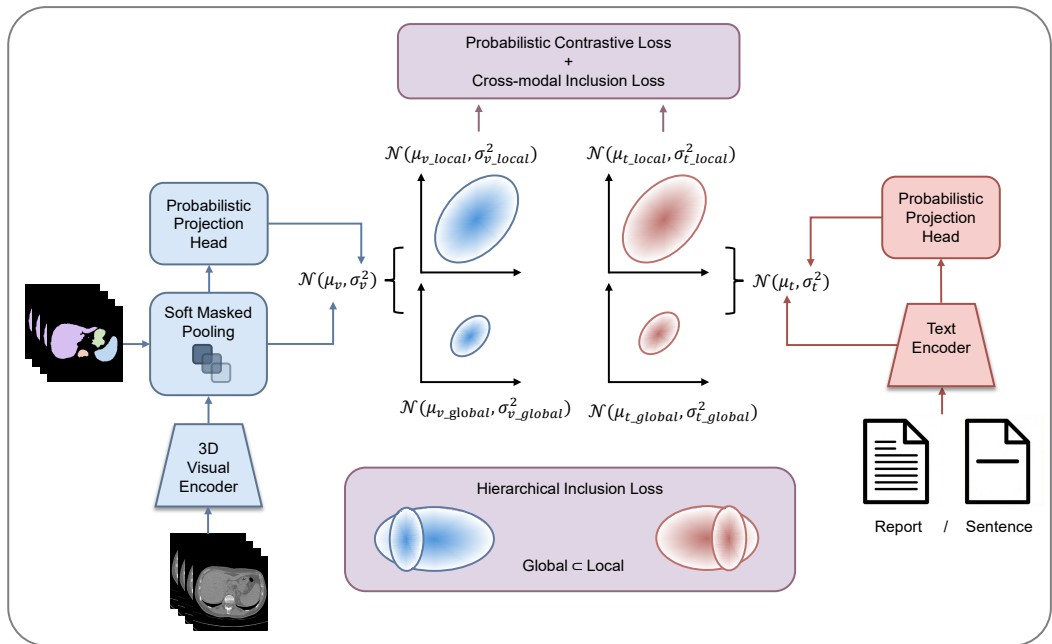

Figure 1: Framework of HiPro-CT

their corresponding sentence descriptions. The entire model is optimized end-to-end using a hybrid objective that combines Probabilistic Contrastive Loss for semantic alignment and Hierarchical Inclusion Loss to enforce geometric constraints between global and local distributions.

### 3.3. Probabilistic Encoders and Uncertainty Estimation

Our framework consists of a 3D visual encoder $f_v$ and a text encoder $f_t$. Instead of deterministic vectors, we model the output embeddings as diagonal Gaussian distributions $p(z|x) = \mathcal{N}(z; \mu(x), \Sigma(x))$, where $\Sigma(x)$ is a diagonal covariance matrix representing the uncertainty.

**3D Visual Encoder** Given a 3D CT volume $V \in \mathbb{R}^{D \times H \times W}$, we employ a 3D Vision Transformer (ViT) (Dosovitskiy, 2020) as the backbone. The input volume is divided into non-overlapping patches and linearly projected into patch embeddings $E_v = \{e_1, e_2, \ldots, e_N\}$, where $N$ is the number of patches.

To estimate the distribution parameters $(\mu_v, \Sigma_v)$, we utilize a specific **Probabilistic Projection Head**:

- **Mean ($\mu_v$):** We perform global average pooling on $E_v$ followed by a linear projection and $L_2$ normalization to map the representation onto a hypersphere:

$$\mu_v = \text{Normalize}(\text{Linear}(\text{MeanPool}(E_v))) \tag{1}$$

- **Variance ($\Sigma_v$):** Unlike previous works that use a static token, we introduce a learnable **Variance Query** $Q_{var} \in \mathbb{R}^{1 \times C}$ to dynamically aggregate uncertainty information from the patch features via a Multi-Head Attention (MHA) mechanism:

$$h_{var} = \text{MHA}(query = Q_{var}, key = E_v, value = E_v) \tag{2}$$

$$\log \sigma_v^2 = \text{Linear}_{var}(h_{var}) \tag{3}$$

where $\Sigma_v = \text{diag}(\exp(\log \sigma_v^2))$. This query-based mechanism allows the model to attend to specific ambiguous regions (e.g., blurry boundaries) when estimating the global uncertainty.

**Text Encoder** Similarly, for the input text $T$, we use a Transformer-based encoder. The global text distribution $\mathcal{N}(\mu_t, \Sigma_t)$ is derived from the last hidden states using a parallel mechanism: the mean is derived from the [CLS] token, and the variance is computed using a learnable text variance query attending to the sequence outputs.

## 3.4. Fine-grained Alignment via Soft Masked Pooling

A critical limitation of standard CLIP on 3D data is the dominance of background information. To enforce fine-grained alignment, we utilize anatomical segmentation masks. However, since masks are pixel-level and ViT features are patch-level, hard cropping leads to information loss. We propose soft masked pooling.

Let $M \in \{0, 1\}^{D \times H \times W}$ be the binary mask for a specific organ. We first downsample $M$ to match the patch resolution of the ViT output, resulting in a soft weight map $W \in [0, 1]^N$, where $W_i$ represents the proportion of the organ within the $i$-th patch. The organ-specific visual mean $\mu_{v\_local}$ is computed as:

$$\tilde{w}_i = \frac{W_i}{\sum_{j=1}^{N} W_j + \epsilon}, \quad \mu_{v\_local} = \text{Normalize}\left(\text{Linear}\left(\sum_{i=1}^{N} \tilde{w}_i \cdot e_i\right)\right) \tag{4}$$

For the organ-specific variance $\Sigma_{v\_local}$, we apply the Variance Query $Q_{var}$ to the masked patch features, ensuring the uncertainty estimate focuses only on the relevant anatomical region. The corresponding local text embedding $\mathcal{N}(\mu_{t\_local}, \Sigma_{t\_local})$ is encoded from the specific sentence describing that organ.

## 3.5. Optimization Objectives

We optimize the network using a hybrid loss function that governs both alignment and distributional relationships.

### 3.5.1. PROBABILISTIC PAIRWISE CONTRASTIVE LOSS (PPCL)

We employ the Closed-form Sampled Distance (CSD) (Chun, 2023) to measure the similarity between two distributions. For a vision distribution $z_v$ and text distribution $z_t$:

$$\text{CSD}(z_v, z_t) = \|\mu_v - \mu_t\|_2^2 + \text{Tr}(\Sigma_v + \Sigma_t) \tag{5}$$

Following SigLIP (Zhai et al., 2023), we implement the contrastive loss using a sigmoid-based formulation. The logit for a pair is defined as:

$$s(z_v, z_t) = a \cdot (\mu_v^\top \mu_t - \frac{1}{2}\text{Tr}(\Sigma_v + \Sigma_t)) + b \tag{6}$$

where $a$ and $b$ are learnable scale and bias parameters. We minimize the binary cross-entropy loss $\mathcal{L}_{PPCL}$ on these logits for both global-global pairs and local-local (organ-sentence) pairs.

### 3.5.2. Hierarchical Inclusion Loss

To model the probabilistic relationship between global information and local information, we introduce a **Hierarchical Inclusion Loss**. We model the hierarchical relationship based on information certainty, a principle applicable to both visual and textual modalities. Global inputs (e.g., the full CT volume or the complete radiology report) provide comprehensive context that resolves semantic ambiguity, yielding more deterministic representations with lower variance. Conversely, local fragments (e.g., organ patches or isolated sentences) often lack boundary context or specific references, leading to inherent uncertainty and broader distributions. Therefore, we enforce a constraint where the concentrated global distribution is probabilistically contained within the diffuse local distribution ($z_{global} \subset z_{local}$). We define an inclusion score $H(z_{global} \subset z_{local})$ based on the log-integral of the probability density functions:

$$H(z_1 \subset z_2) = \log \int p_1^2(x)p_2(x)dx - \log \int p_1(x)p_2^2(x)dx \tag{7}$$

A positive $H$ indicates that $z_1$ is likely included in $z_2$. We apply this loss to enforce:

(1) Visual Inclusion: Global CT distribution $\subset$ Organ CT distribution.

(2) Textual Inclusion: Full report distribution $\subset$ Sentence distribution.

The loss is formulated as $\mathcal{L}_{hier} = -\log \sigma(H(z_{global} \subset z_{local}))$.

### 3.5.3. Cross-modal Inclusion Loss

Additionally, following the intuition that text descriptions are often more abstract (and thus more uncertain) than specific images, we apply a cross-modal inclusion loss $\mathcal{L}_{cross}$ to encourage the text distribution to encompass the image distribution (Image $\subset$ Text), further regularizing the uncertainty estimation.

### 3.5.4. Total Objective

Finally, to prevent variance collapse, we add a Variational Information Bottleneck (VIB) regularization term $\mathcal{L}_{vib}$:

$$\mathcal{L}_{vib} = KL(\mathcal{N}(\mu, \Sigma)\|\mathcal{N}(0, I)) = \frac{1}{2}\sum_{i=1}^{D}(\mu_i^2 + \sigma_i^2 - 1 - \log\sigma_i^2) \tag{8}$$

where KL is the Kullback-Leibler divergence. The total objective is:

$$\mathcal{L} = \mathcal{L}_{PPCL} + \lambda_1\mathcal{L}_{hier} + \lambda_2\mathcal{L}_{cross} + \lambda_3\mathcal{L}_{vib} \tag{9}$$

where $\lambda_1$, $\lambda_2$, $\lambda_3$ are hyperparameters balancing the contributions. This multi-objective optimization ensures that our model learns discriminative features for zero-shot classification while maintaining a structured, interpretable probabilistic embedding space.

## 4. Experiments and Results

### 4.1. Implementation Details

The 3D visual encoder consists of a 6-layer ViT trained from scratch, and the text encoder is initialized with the pre-trained BiomedVLP-CXR-BERT-specialized weights (Boecking et al., 2022). Input CT volumes are standardized to a resolution of $336 \times 336 \times 96$ via center cropping or padding, with a patch size of $16 \times 16 \times 8$. During training, each sample consists of a global volume-report pair. Additionally, we employ a stochastic sampling strategy that randomly selects one organ mask and its corresponding sentence per step to compute the fine-grained inclusion loss. The model is trained for 50,000 steps on 4 NVIDIA H800 GPUs with a per-GPU batch size of 4. We optimize the network using the AdamW optimizer (Loshchilov and Hutter, 2017) with a learning rate of $1 \times 10^{-5}$ and a cosine decay scheduler, utilizing bfloat16 mixed-precision to enhance efficiency. Finally, the loss balancing coefficients $\lambda_1$, $\lambda_2$, and $\lambda_3$ are set to 0.1, 0.0001, and 0.1. All experimental results reported in the following tables are averaged over 1,000 bootstrap runs on the test set, with 95% confidence intervals shown in parentheses.

### 4.2. Zero-shot multi-abnormality detection

To evaluate the generalization capability and clinical validity of the learned representations, we conducted zero-shot multi-abnormality detection on the RadGenome-Chest test set, with ground truth labels derived from the original CT-RATE annotations. We compared our framework against state-of-the-art baselines including CT-Net (Draelos et al., 2021) and CT-CLIP. Critically, to ensure a fair comparison that isolates the contribution of architectural design from external pre-training benefits, we re-initialized the visual encoder of the CT-CLIP baseline and retrained it from scratch using identical hyperparameters and sampling strategies as HiPro-CT. During inference, we employed the text prompts "{*Abnormality*} *is present*" and "{*Abnormality*} *is not present*" to compute similarity scores, and as shown in Table 1, our proposed probabilistic framework significantly outperforms the standard CLIP paradigm trained under the same conditions, demonstrating superior zero-shot transferability.

### 4.3. Cross-modal Retrieval

To further evaluate the granularity of semantic alignment beyond simple classification, we conducted cross-modal retrieval experiments on a randomly sampled subset of 100 volume-report pairs from the test set. We performed both Text-to-Image (T2I) and Image-to-Text (I2T) retrieval, reporting Recall@K (R@1, R@5, R@10) metrics. For the deterministic CT-CLIP baseline, rankings were generated using standard cosine similarity. In contrast, HiPro-CT utilizes the negative Closed-form Sampled Distance to rank candidates, leveraging the learned variance to measure probabilistic overlap. As detailed in Table 2, HiPro-CT consistently outperforms the baseline. This superiority suggests that our probabilistic

framework effectively captures distinct pathological details via soft masked pooling and handles the semantic ambiguity of reports better than point-based embeddings, resulting in more precise matching in the open retrieval space.

Table 1: Zero-shot multi-abnormality detection on four metrics: Accuracy, Precision, F1 Score (Weighted), AUROC

| Method | Type | Accuracy | Precision | F1 Score | AUROC |
|---|---|---|---|---|---|
| CT-Net | Supervised | 0.617 (0.609, 0.624) | 0.264 (0.257, 0.272) | 0.657 (0.646, 0.667) | 0.629 (0.620, 0.638) |
| ViT-3D | Supervised | **0.815** (0.809, 0.822) | 0.235 (0.223, 0.248) | **0.777** (0.770, 0.785) | 0.709 (0.697, 0.719) |
| CT-CLIP | Zero-shot | 0.643 (0.632, 0.653) | 0.290 (0.279, 0.301) | 0.680 (0.670, 0.690) | 0.679 (0.668, 0.689) |
| HiPro-CT | Zero-shot | 0.684 (0.670, 0.695) | **0.326** (0.315, 0.337) | 0.716 (0.704, 0.727) | **0.729** (0.720, 0.738) |

Table 2: Comparison of retrieval accuracy between CT-CLIP and HiPro-CT. Results are reported as Recall@K (%) on the test subset ($N = 100$).

| Method | Image-to-Text | | | Text-to-Image | | |
|---|---|---|---|---|---|---|
| | **R@1** | **R@5** | **R@10** | **R@1** | **R@5** | **R@10** |
| CT-CLIP | 9.77 (8.82, 10.74) | 33.06 (31.97, 34.14) | 49.77 (48.72, 50.77) | 9.95 (9.08, 10.87) | 33.20 (32.16, 34.27) | 49.85 (48.85, 50.83) |
| HiPro-CT | **10.05** (9.08, 11.06) | **36.13** (34.97, 37.28) | **54.55** (53.52, 55.56) | **10.26** (9.27, 11.25) | **35.66** (34.40, 36.83) | **54.69** (53.64, 55.75) |

### 4.4. Ablation Study on Hierarchical Probabilistic Constraints

To investigate the individual contributions of our proposed optimization objectives, we conducted a progressive ablation study with three configurations: (1) using solely the Global Probabilistic Pairwise Contrastive Loss (PPCL) as a baseline; (2) combining Global and Local PPCL to introduce fine-grained organ-level supervision; and (3) the full HiPro-CT framework which further incorporates Hierarchical and Cross-modal Inclusion Losses. Importantly, this experimental design also provides a direct ablation on replacing the standard deterministic CLIP objective with our probabilistic formulation. Specifically, the CT-CLIP baseline in Table 1 serves as the standard point-embedding CLIP loss benchmark (retrained from scratch with identical hyperparameters; see Section 4.2), while "PPCL Loss (Global)" in Table 3 applies PPCL only at the global level (distributional embeddings) without any local (mask/organ-level) alignment or hierarchical inclusion constraints. Their comparison

therefore isolates the effect of mapping inputs to distributions, which improves zero-shot detection, whereas for cross-modal retrieval the global-only probabilistic objective is not uniformly better (see Table 2 vs. Table 4), motivating the added fine-grained and hierarchical constraints.

Table 3: Ablation study on zero-shot multi-abnormality detection performance across different loss configurations.

| Loss Function | Accuracy | Precision | F1 Score | AUROC |
|---|---|---|---|---|
| PPCL (Global) | 0.655 (0.643, 0.668) | 0.303 (0.291, 0.314) | 0.691 (0.680, 0.702) | 0.702 (0.692, 0.713) |
| PPCL (Global & Local) | 0.673 (0.660, 0.685) | 0.325 (0.315, 0.337) | 0.707 (0.695, 0.718) | 0.720 (0.711, 0.730) |
| PPCL (Global & Local) $+ L_{\text{hier}}$ | 0.679 (0.665, 0.692) | 0.321 (0.310, 0.333) | 0.711 (0.699, 0.722) | 0.724 (0.715, 0.734) |
| PPCL (Global & Local) $+ L_{\text{hier}} + L_{\text{cross}}$ | 0.681 (0.668, 0.693) | **0.327** (0.316, 0.338) | 0.713 (0.701, 0.723) | 0.726 (0.717, 0.736) |
| PPCL (Global & Local) $+ L_{\text{hier}} + L_{\text{cross}} + L_{\text{vib}}$ | **0.684** (0.670, 0.695) | 0.326 (0.315, 0.337) | **0.716** (0.704, 0.727) | **0.729** (0.720, 0.738) |

Table 4: Ablation study on cross-modal retrieval performance across different loss configurations.

| Loss Function | Image-to-Text Retrieval | | | Text-to-Image Retrieval | | |
|---|---|---|---|---|---|---|
| | R@1 | R@5 | R@10 | R@1 | R@5 | R@10 |
| PPCL (Global) | 8.62 (7.74, 9.59) | 30.54 (29.35, 31.71) | 47.80 (46.87, 48.85) | 9.01 (8.06, 10.04) | 32.23 (31.14, 33.31) | 48.56 (47.51, 49.62) |
| PPCL (Global & Local) | 10.04 (9.08, 11.06) | 33.80 (32.67, 34.85) | 51.50 (50.45, 52.56) | 9.90 (8.95, 10.87) | 33.89 (32.74, 35.04) | 51.89 (50.90, 52.88) |
| PPCL (Global & Local) $+ L_{\text{hier}}$ | 10.10 (9.08, 11.10) | 34.96 (33.82, 36.13) | 53.06 (52.05, 54.16) | **10.64** (9.65, 11.64) | 35.10 (33.89, 36.25) | 53.28 (52.30, 54.28) |
| PPCL (Global & Local) $+ L_{\text{hier}} + L_{\text{cross}}$ | **10.12** (9.08, 11.13) | 35.56 (34.34, 36.77) | 54.17 (53.13, 55.18) | 10.32 (9.27, 11.32) | **36.16** (34.97, 37.34) | 54.40 (53.32, 55.56) |
| PPCL (Global & Local) $+ L_{\text{hier}} + L_{\text{cross}} + L_{\text{vib}}$ | 10.05 (9.08, 11.06) | **36.13** (34.97, 37.28) | **54.55** (53.52, 55.56) | 10.26 (9.27, 11.25) | 35.66 (34.40, 36.83) | **54.69** (53.64, 55.75) |

As presented in Table 3 and Table 4, the quantitative results exhibit a clear monotonic improvement across all metrics as the loss functions become more comprehensive. Specifically, the introduction of Local PPCL significantly outperforms the global-only baseline by mitigating feature dilution, while the final integration of inclusion losses yields the best performance by explicitly enforcing geometric logical constraints within the probabilistic space. Detailed results are presented in Appendix A. This consistent upward trend validates our hypothesis that modeling both multi-granularity semantics and their hierarchical interrelations is essential for robust 3D medical vision-language alignment.

### 4.5. Qualitative Analysis

To complement the quantitative evaluation, we provide two qualitative analyses in the appendix. Appendix B visualizes the learned Gaussian embeddings, confirming that the hierarchical inclusion relationships are preserved across both visual and textual modalities. Appendix C presents Grad-CAM-based localization comparisons, showing that HiPro-CT attends to more clinically relevant regions than CT-CLIP.

## 5. Conclusion

In this paper, we introduced HiPro-CT, a hierarchical probabilistic framework designed to overcome the critical limitations of feature dilution and semantic ambiguity in 3D medical vision-language alignment. By synergizing Gaussian distributional embeddings with a novel Soft Masked Pooling mechanism, our approach enables precise organ-level alignment while explicitly modeling the inherent uncertainty of clinical data through variance and hierarchical geometric constraints. Extensive experiments on the RadGenome-Chest CT dataset demonstrate that HiPro-CT achieves superior performance over state-of-the-art deterministic baselines in both zero-shot abnormality detection and cross-modal retrieval. These results collectively validate the efficacy of integrating fine-grained anatomical supervision with probabilistic representation learning, offering a robust and interpretable paradigm for the advancement of 3D medical foundation models.

## Acknowledgments

This work was supported by the Institute for Intelligent Healthcare, Tsinghua University (No. 2022ZLB001) and the Tsinghua-Foshan Innovation Special Fund (No. 2021THFS0104).

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

## Appendix A. Results of zero-shot multi-abnormality detection

Compared to CT-CLIP, HiPro-CT demonstrates comprehensive superiority across all four evaluation metrics (Accuracy, Precision, F1 Score, and AUROC) in the detection of the vast majority of anomalies, including medical material, arterial and coronary artery wall calcification, cardiomegaly, pericardial and pleural effusion, lymphadenopathy, atelectasis, lung opacity, pulmonary fibrotic sequela, mosaic attenuation pattern, peribronchial and interlobular septal thickening, and consolidation. In contrast, CT-CLIP consistently outperforms HiPro-CT across all metrics in the detection of lung nodules and bronchiectasis.

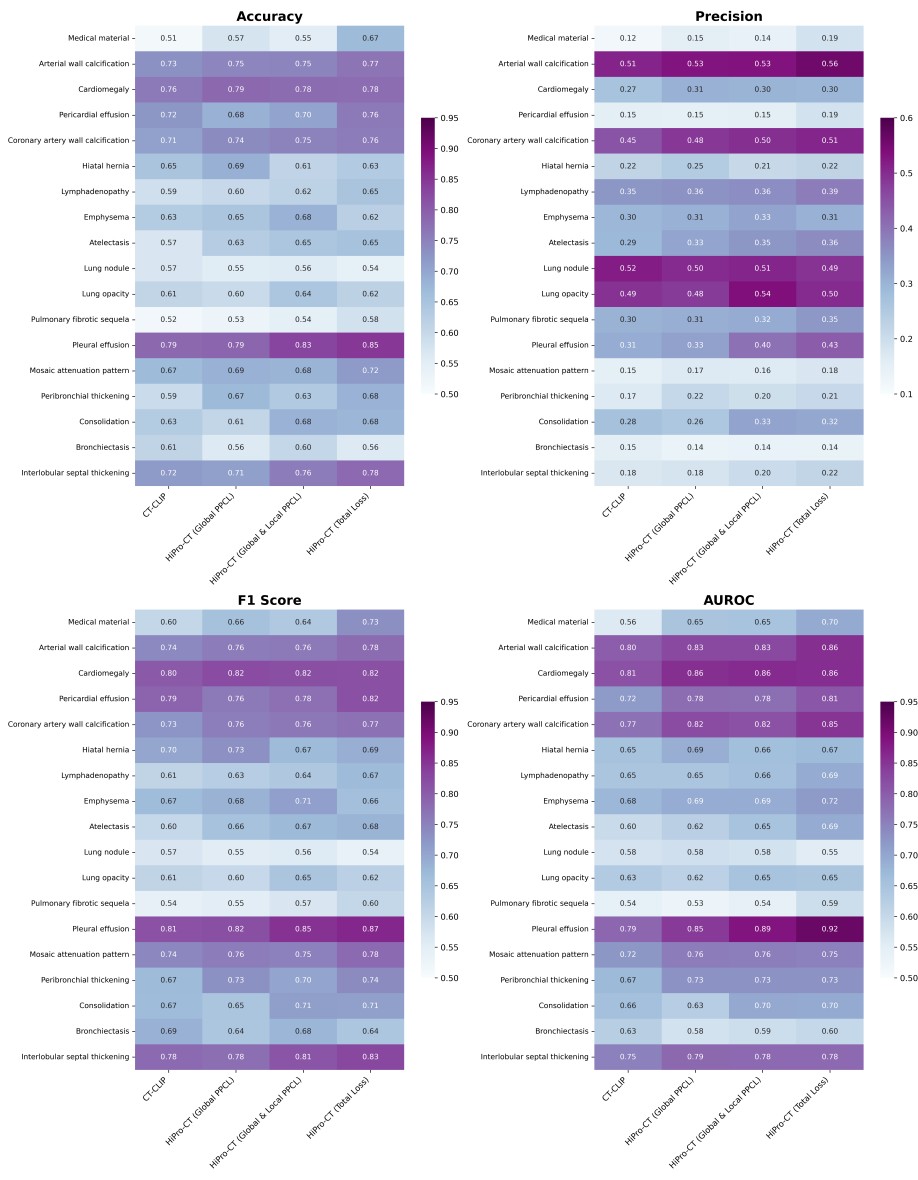

Figure 2: Heatmap of zero-shot multi-abnormality detection results

## Appendix B. Visualizing Distributions

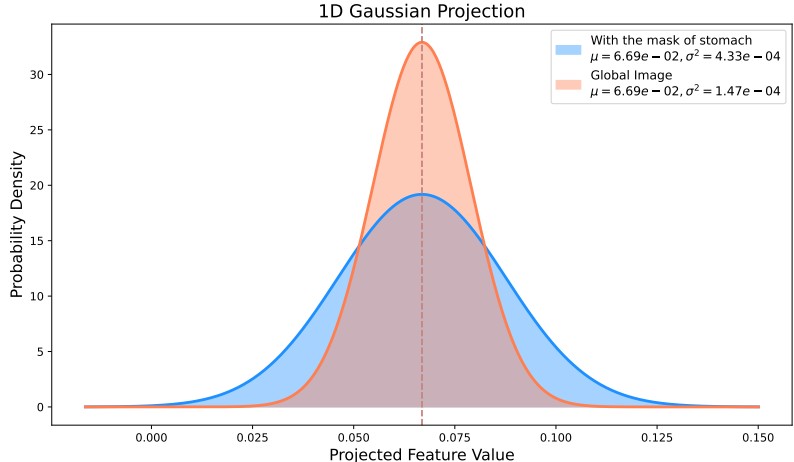

Figure 3: Gaussian projection visualization for image embeddings.

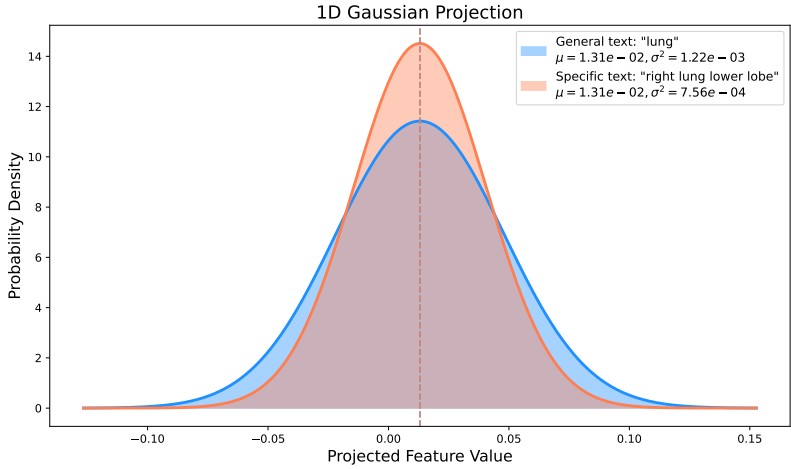

Figure 4: Gaussian projection visualization for text embeddings.

These visualization results further corroborate the core intuition underlying our hierarchical probabilistic constraints. Specifically, to facilitate visualization, we project the original multi-dimensional Gaussian embeddings onto a one-dimensional Gaussian distribution. By selecting a projection axis that aligns the projected means as closely as possible, we can directly inspect the resulting variances and their inclusion relationships.

As illustrated in Figure 3, the Gaussian embedding derived from the masked input (e.g., utilizing a stomach mask to perform soft masked pooling) yields a local distribution that statistically encompasses the distribution encoded by the corresponding global volume. This phenomenon indicates that local anatomical semantics entail higher information un-

certainty; conversely, as global information is acquired, the probability distribution function becomes notably more concentrated.

Similarly, Figure 4 demonstrates the textual hierarchy: the distribution of a shorter, more abstract description (e.g., "lung") encompasses that of a longer, more specific one (e.g., "right lung lower lobe"). Collectively, these results validate that our learned probabilistic space preserves hierarchical inclusion relationships across both visual and textual embeddings.

## Appendix C. Localization results

To qualitatively assess localization behavior, we visualize model attention using Grad-CAM (Selvaraju et al., 2020). As shown in Figure 5, HiPro-CT exhibits more precise and clinically relevant focus on fine-grained regions corresponding to the target findings, whereas CT-CLIP tends to produce more diffuse or partially misplaced activations. This comparison suggests that the proposed fine-grained supervision together with probabilistic hierarchical constraints improves the model's ability to attend to the correct anatomical details.

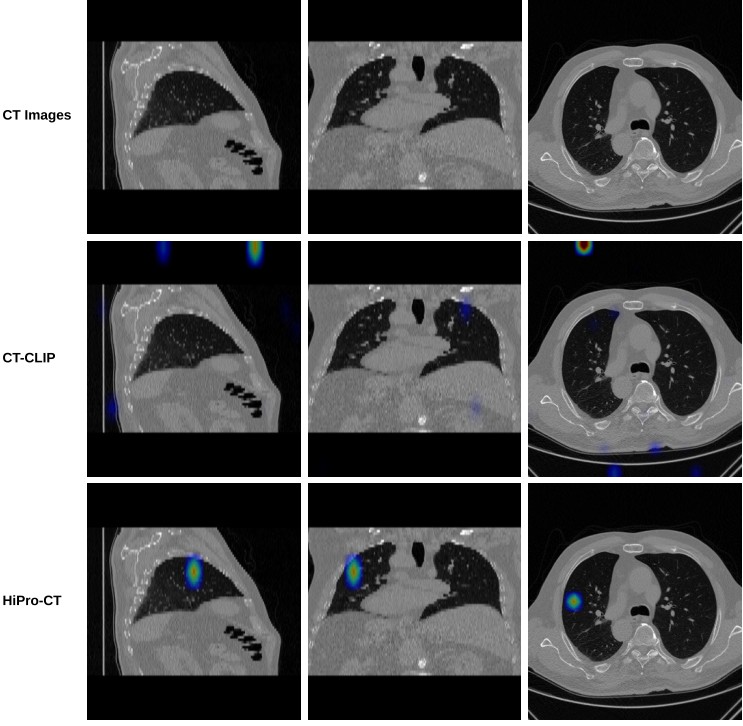

Figure 5: Localization result for the text: *"In the left lung, there is linear density consistent with band atelectasis-sequelae changes in the inferior lingular segment."*

