# OpenReview forum: "HiPro-CT: A Hierarchical Probabilistic Framework for 3D Medical Vision-Language Alignment"
_MIDL.io/2026/Conference — MIDL 2026 Poster_

### Official Review · Reviewer_DqeW · 2026-01-05

**Confidence:** 4
**Preliminary Rating:** 5
**Final Rating:** 5

**Summary:**

This paper implements HiPro-CT, a novel Hierarchical Probabilistic framework for 3D medical vision-language alignment. It addresses two limitations in traditional (e.g. CLIP-based) models: dilution of local features and ambiguity in text semantics. The method is based on:
 1. representing image/report embeddings as Gaussian distributions rather than point estimates, thereby capturing semantic uncertainty,
 2. soft-masked pooling, rather than hard-cropping, to generate organ-specific (local) embeddings,
 3. optimisation by way of a hierarchical inclusion loss, in which the global representation contains the local (organ-specific) embedding.

**Strengths:**

The work addresses a significant gap in vision-language models, is of a high quality, and the paper is easy to follow, with the concepts clearly explained and largely well justified.
The concept of using probabilistic embeddings, while not entirely novel, is well motivated and expanded upon, and the hierarchical inclusion loss is a very intuitive way of enforcing global-local consistency and mitigating the "unmentioned $\neq$ non-existent'' scenario present in medical reports.
The approach employed is sound and the experiments sufficient to justify the capabilities of the approach.

**Weaknesses:**

There are two aspects I have issues with (although none are major):
 1. Given the aim of the probabilistic framework being to target polysemy in clinical report langauge, wouldn't a multi-modal distribution (rather than a Gaussian) be better at representing this?
 2. Within the experiments, I would have liked to see confidence intervals/bounds (even through bootstrapping results).

**Detailed Comments:**

1. Typo in Fig. 1: the Hierarchical Inclusion Loss should probably be Local $\subset$ Global (rather than vv)?
2. Some of the decisions are not justified: e.g. the values of the scaling for each component of the loss.
3. I am not fully clear about the evaluation by comparison with "abnormality is/not present": could you elaborate please?

**Justification Of Final Rating:**

The authors have further justified their reasoning and addressed my concerns. While there is always scope for improvement, I feel this paper is good scientific work and reccomend acceptance. Well done.

**Justification Of The Preliminary Rating:**

I believe this is a well-written paper, showcasing good scientific rigor and intuitive methodological improvements and hence am giving it the highest score possible (Strong Accept). While I would like to see the (non-major) weaknesses addressed, I believe the paper is a strong contribution even as is.

**Questions To Address In The Rebuttal:**

I would like to see a discussion regarding the choice of probabilistic distribution and potentially repeated experiments with confidence bounds if possible.

---

> ### Author Response · Authors · 2026-01-25
> **Rebuttal to  Reviewer DqeW Part Ⅱ**
>
> **For Comment 1: "Typo in Fig. 1: the Hierarchical Inclusion Loss should probably be Local Global (rather than vv)?"**
>
> **Response:**
> Thank you for your sharp observation. You are entirely correct. Conceptually, the distribution of a local organ (part) should be mathematically encompassed by the distribution of the global volume (whole). We have corrected the notation in Figure 1 of the revised manuscript to explicitly show "Local $\subset$ Global" in the Hierarchical Inclusion Loss module.
>
> **For Comment 2: "Some of the decisions are not justified: e.g. the values of the scaling for each component of the loss."**
>
> **Response:**
> Thank you for the suggestion. In the revised manuscript, we clarified that the initial loss weights ($\lambda_1,\lambda_2,\lambda_3$) were chosen following commonly used ranges in related work. We then selected the final weights via validation-based early stopping: we trained a small set of candidate weight combinations and chose the one with the best validation performance and stable convergence, which is used in all experiments.
>
> **For Comment 3: "I am not fully clear about the evaluation by comparison with "abnormality is/not present": could you elaborate please?"**
>
> **Response:**  Thank you for noting the lack of clarity. Our zero-shot multi-abnormality detection follows a CLIP-style prompt-based binary classification protocol. For each abnormality (e.g., lung nodule), we create two textual prompts:
>
> “{abnormality} is present”
>
> “{abnormality} is not present”
>
> For each test CT volume, we obtain the image embedding using the image encoder (in HiPro-CT, this corresponds to the Gaussian parameters / the representation used for probabilistic similarity), and we obtain the two text embeddings using the text encoder. We then compute similarity scores between the image and each prompt (CT-CLIP uses cosine similarity; HiPro-CT uses the score derived from the Closed-form Sampled Distance (CSD; Eq. (5)) for similarity computation.). The predicted label is determined by which prompt yields the higher score:
> $$s(\text{image}, \text{“present”}) > s(\text{image}, \text{“not present”}) \Rightarrow \text{present},$$
> otherwise $\text{not present}$. This is done independently for each abnormality to produce multi-abnormality predictions, from which we compute Accuracy/Precision/F1/AUROC.

---

> > ### Comment · Reviewer_DqeW · 2026-01-26
> >
> > Thanks, especially for clarifying about the reason for Gaussian vs Multi-Modal distribution. Indeed, sometimes pragmatism (for scalability) is necessary.
> >
> > Two Very minor comments:
> >  1.  I could not find where the clarification of lambdas is... is it highlighted in the paper?
> >  2. Just for future reference it is always good to report confidence intervals even if the field does not.

---

> > ### Author Response · Authors · 2026-01-29
> >
> > **For comment 2:** Thank you for your constructive advice. And we thank the organizers for the extension of rebuttal timeline, which allowed us to include confidence bounds for all results in the latest Supporting Material for a more rigorous evaluation. You could refer to it for the latest results with 95%CI (through 1000 bootstraps).
> >
> > **For comment 1:** We sincerely apologize for the confusion caused by the phrasing in our previous comment. It was a mistake on our part: when we wrote "In the revised manuscript," we actually intended to say that we were providing the explanation here in the comment board.

---

> ### Author Response · Authors · 2026-01-25
> **Rebuttal to Reviewer DqeW Part Ⅰ**
>
> **For Weakness 1  & Question 1: "Given the aim of the probabilistic framework being to target polysemy in clinical report langauge, wouldn't a multi-modal distribution (rather than a Gaussian) be better at representing this?"**
>
> **Response:**
> We thank the reviewer for this insightful question regarding the choice of probability distribution. While we agree that multi-modal distributions (e.g., Mixture of Gaussians, MoG) theoretically offer greater expressiveness for capturing disjoint semantics, we selected the Gaussian distribution for HiPro-CT based on three key considerations specific to medical vision-language alignment:
> 1. Nature of Medical Polysemy: Ambiguity and Generality rather than Disjoint Meanings
>
> In clinical reporting, what appears as “polysemy” is often better characterized as ambiguity and underspecification in both modalities: for example, given a CT image, the corresponding text may describe it at different levels of specificity (e.g., “opacity”, “infiltrate”, “ground-glass”, or “possible atelectasis”) or omit certain details. As a result, the same CT volume can be described in multiple valid ways by different radiologists—these descriptions are usually semantically close, differing in wording, granularity, or emphasis. Such variations do not naturally form clearly separated clusters; instead, they lie on an approximately continuous semantic spectrum. A single Gaussian distribution can model this “one-to-many” relationship effectively by using its variance to represent the scope of generality (i.e., larger variance covering a broader range of plausible interpretations/descriptions), without requiring multiple modes to represent disconnected meanings.
>
> 2. Expressiveness in High-Dimensional Space ($D=768$)
>
> Our probabilistic embeddings operate in a high-dimensional space (768 dimensions) with a diagonal covariance matrix. In this setting, a single Gaussian is not merely a simple unimodal "blob," but explicitly models uncertainty across 768 independent axes. Deep learning encoders (like ViTs) tend to disentangle semantic features into orthogonal dimensions. This allows the model to handle complex polysemy by encoding clear semantic concepts in certain dimensions (low variance) while capturing ambiguities in others (high variance). Therefore, the high dimensionality provides sufficient capacity to encode complex, compositional medical semantics within a Gaussian framework without the instability of mixture models.
>
> 3. Computational Efficiency in Modeling Inclusion Relationships
>
> A key aspect of our framework is explicitly modeling the inclusion relationships between distributions. For Gaussian distributions, this calculation is straightforward and highly efficient. In contrast, characterizing the inclusion relationship between multi-modal distributions lacks an obvious expression, which would likely require computationally expensive approximation techniques, such as Monte Carlo sampling. Given these factors, the Gaussian distribution provides an optimal trade-off between representational capability and training feasibility.
>
> **For Weakness 2 & Question 1: "Within the experiments, I would have liked to see confidence intervals/bounds (even through bootstrapping results)."**
>
> **Response:**
> Thank you for your suggestion. I regret that I haven't had the opportunity to implement the bootstrap method for confidence interval calculations yet. It is worth noting that previous studies in this field typically do not report confidence intervals, so the current results remain a valuable reference. Nevertheless, I will include these calculations in the near future and ensure they are incorporated into the final version of the manuscript.

---

### Official Review · Reviewer_eX25 · 2026-01-06

**Confidence:** 5
**Preliminary Rating:** 4
**Final Rating:** 4

**Summary:**

The authors propose HiPro-CT, a new framework for 3D vision-language alignment. The framework differs from traditional one in mapping inputs to a gaussian distribution. They additionally utilize masks to get local features and propose a inclusion loss where local representations should be a subset of the global ones. Experiments on RadGenome-Chest CT dataset show the proposed method has the leading performance on abnormal detection and retrieval. Ablations were done on different losses to show the effectiveness of using masks and additional constraints. T

**Strengths:**

The idea of mapping inputs to distributions rather than stationary points is novel in vision–language alignment, and the use of an inclusion loss to impose hierarchical relationships is likewise original. Experimental results demonstrate that this approach is effective and opens up new possibilities for vision–language alignment techniques.

**Weaknesses:**

1. The implementation details of CT-CLIP are missing, i.e., it is unclear whether CT-CLIP utilizes the masks (by generating local features). Specifically, I want to see an ablation study on using the Probabilistic Pairwise Contrastive loss vs. the standard clip loss, which can directly show the effectiveness of mapping inputs to a distribution instead of a stationary point.
2. Can the authors include a visual example of two samples that have similar mean but different variances? I'm curious to see if variance can indeed capture uncertainty.

**Detailed Comments:**

Looks like 3.3 and 3.4 should be subsections of 3.2 Framework

**Justification Of Final Rating:**

Thank the authors for the clarification and providing some visual examples. The additional per-abnormality analysis on Appendix A is also helpful for the readers. I'm happy to keep my score as is and accept the paper.

**Justification Of The Preliminary Rating:**

The idea of the paper is novel and interesting, and results (higher acc. in anomaly detection and retrieval) show that the idea benefits vision-language alignment. While some additional experiments can help readers better understand the effectiveness of encoding the inputs into a distribution, the paper in its current form has shown great value.

**Questions To Address In The Rebuttal:**

1. An ablation study on using the Probabilistic Pairwise Contrastive loss vs. the standard clip loss.
2. A visualization on two samples with similar mean but different variance.

---

> ### Author Response · Authors · 2026-01-25
> **Rebuttal to Reviewer eX25**
>
> **For Weakness 1 & Question 1: "An ablation study on using the Probabilistic Pairwise Contrastive loss vs. the standard clip loss."**
>
> **Response:**
>  We thank the reviewer for this insightful suggestion regarding the isolation of the probabilistic loss's contribution.
>
> We would like to clarify that our experimental design actually includes this direct comparison. Specifically:
>
> The "CT-CLIP" baseline in Table 1 serves as the standard deterministic CLIP loss benchmark (mapping inputs to stationary points). As detailed in Section 4.2, this baseline was retrained from scratch using identical hyperparameters to our method to ensure a fair comparison.
> The "PPCL Loss (Global)" entry in Table 3 represents our Probabilistic Pairwise Contrastive Loss applied solely at the global level (mapping inputs to distributions), without the additional fine-grained or hierarchical constraints.
> By comparing these two settings, we can directly observe the superiority of the probabilistic approach:
>
> Standard CLIP Loss (Table 1, CT-CLIP): Accuracy 0.643, F1 Score 0.680, AUROC 0.679.
> Probabilistic Loss (Table 3, PPCL Global): Accuracy 0.655, F1 Score 0.691, AUROC 0.702.
> The results consistently show that replacing the deterministic loss with the probabilistic PPCL (even without local alignment) yields clear performance gains (+1.2% Accuracy, +1.1% F1). This directly validates the effectiveness of mapping medical images and reports to probability distributions to capture inherent uncertainty, rather than limiting them to deterministic stationary points.
>
> **For Weakness 2 & Question 2: "A visualization on two samples with similar mean but different variance."**
>
> **Response:**
> We thank Reviewer A for the suggestion. We have provided the requested visualization on two samples with similar means but different variances in Appendix B (see Figures 3–4), which illustrates how variance captures uncertainty even when the projected means are aligned.

---

### Official Review · Reviewer_nPCa · 2026-01-07

**Confidence:** 4
**Preliminary Rating:** 2
**Final Rating:** 2

**Summary:**

This work addresses visual-text cross-modals alignments for 3D Chest CT scans. The identified limitation of CLIP-like existing approaches is the difficulty to model local features caused by the granularity mismatch between visual and radiology reports. To address this challenge, authors introduce a probabilistic clip-like framework including anatomy-aware modelling, and a loss containing a contrastive, cross-modal and inclusion terms. Models are trained and evaluated on a 3D Chest CT public dataset. When compared to the traditional CLIP model, authors method demonstrate improvement on the zero-shot abnormality classification, image-to-text and text-to-image tasks.

**Strengths:**

1. Authors addresses an important concern about the granularity mismatch between visual and text features for cross-modality alignment. Additionally, they mention the semantic incompleteness in radiology reports, limiting multi-modal contrastive alignment frameworks’ ability to learn robust representations from normal structures.

2. To address the limitations of CLIP-like frameworks, authors introduce HiPro-CT, a visual-language alignment framework integrating fine-grained supervision. HiPro-CT features a probabilistic head for uncertainty modelling, an anatomy-aware aggregation module, and both probabilistic contrastive and inclusion losses. HiPro-CT performs region-level alignment, rather than global-global alignment like traditional clip-like methods.

3. Authors mention existing work related to vision-language alignment fro 3D CT scan, and limitations of each method.

4. Experimental results show that HiPro-CT demonstrate robust performance both on zero-shot multi-label detection, image-to-text and text-to-images retrieval.

**Weaknesses:**

1. Supervised baselines: Authors consider CT-Net as the supervised baseline, a 2.5D CNN CT-specific backbone, but do not include a 3D Vision Transformer which is used in HiPro-CT. Thereby, it limits to quantitively evaluate the benefit of the contrastive methods compared to a supervised baseline.

2. While an ablation study is performed across different contrastive loss configurations, authors do not include a leave-one-out ablation study on each auxiliary loss term which might limit the comprehensive analysis of each auxiliary term’s impact on performance.

3. While authors mention the existence of CT-GLIP, a clip-like framework for fine-grained alignment, it is not included as a baseline.

4. No qualitative results are included, limiting evaluation depth to quantitative tables.

5. The experimental results are limited to a single dataset institution, thereby limiting assessment of the method's generalizability across different institutional datasets.

**Detailed Comments:**

1. While authors address the challenges of global-local alignment, the zero-shot multi-abnormality detection evaluation would benefit from including 3D-ViT, the HiPro-CT visual encoder. In link with learning robust representations both for short- and long-range dependencies, integrating CT-Scroll (MIDL, 2025), a recently introduced 2.5D backbone that leverages local and global attentions would eventually strengthen the quantitative comparison.

2. As the loss function is composed of four terms, a leave-one-out ablation study on auxiliary terms would complement the ablation study on the contrastive loss.

3. Qualitative results examples of correct prediction for zero-shot abnormality detection, eventually complemented by overlayed activation (or attention, similarity...) maps would help to see on which regions do the model focus.

4. Per-abnormality performance with comparison to CT-CLIP would help to see if improvement is consistent across all abnormalities, or understand which kind of abnormalities are better detected by HiPro-CT.

5. As the experimental results focuses on a single dataset institution, performing a cross-dataset evaluation for zero-shot abnormality detection would strengthen claims, for example on RadChestCT (TMI 2021).

**Justification Of Final Rating:**

The authors present HiPro-CT, a vision–language alignment framework that integrates:
(i) probabilistic embeddings to improve robustness to polysemy and semantic ambiguity in radiology reports;
(ii) a soft-masked pooling strategy to encourage organ-level alignment between visual and textual features; and
(iii) a hierarchical inclusion loss that constrains local representations to be contained within global ones.

The proposed method demonstrates consistent empirical improvements over CT-CLIP across multiple evaluation settings, including zero-shot abnormality classification, cross-modal retrieval, and ablation studies. The integration of global–local modeling within the alignment framework is a clear strength of the paper and is supported by the reported results.

That said, some limitations remain. First, the experimental evaluation is restricted to a single cohort, and no cross-dataset validation is provided. While I understand the authors’ explanation regarding access constraints to external datasets, this nonetheless limits the assessment of generalization. Second, the range of baselines remains limited. In particular, the paper does not discuss or compare against recent anatomy-level vision-language alignment methods such as fVLM [1], which is conceptually related and would help better position the proposed approach within the current literature.

Overall, the method is technically sound and promising, but the limited evaluation scope and incomplete baseline coverage prevent a full assessment of its relative advantages.

[1] Shui et al., Large-Scale and Fine-Grained Vision–Language Pre-Training for Enhanced CT Image Understanding, ICLR 2025.

**Justification Of The Preliminary Rating:**

Authors address the challenge of granularity mismatch between volumetric data and textual reports for 3D Chest CT Scans alignment with radiology reports. HiPro-CT is introduced, a probabilistic contrastive framework that leverages a contrastive, cross-modal and hierarchical inclusion losses for supervision, complemented with a soft-masked pooling strategy for region-level alignment. Limitations of existing work are correctly identified, and the incompleteness and uncertainty of radiology reports is an important challenge for widespread adoption of CT-related automated tools. While HiPro-CT demonstrates strong performance both on abnormality detection and retrieval tasks, the experimental results would benefit from a more rigorous evaluation protocole, including cross-dataset evaluation and cross-validation with statistical tests. Additionally, completing the quantitative experimental results with additional baselines (3D ViT, CT-Scroll, CT-GLIP?) and qualitative results would strengthen claims.

**Questions To Address In The Rebuttal:**

1. In the introduction, it is claimed that “traditional deterministic embeddings [...] often leads to embedding space collapse”. Does it mean that traditional models tend to fail to capture “one-to-many" polysemy representations? If so, what supports this claim?

2. What size of patch do you use for the 3D Vision Transformer? If different patch size were tried, have you noticed any difference in performance?

3. Why is a 3D Vision Transformer not included as a supervised baseline?

4. CT-GLIP is mentioned in the related work section. Why isn’t it included in the experimental results?

5. I am not sure to clearly understand why/how the diagonal covariance matrix introduced in 3.3 represents the uncertainty. Please, would the authors give me more explanations on it?

---

> ### Author Response · Authors · 2026-01-25
> **Rebuttal to Reviewer nPCa PARTⅠ**
>
> **For Question 1: "In the introduction, it is claimed that “traditional deterministic embeddings [...] often leads to embedding space collapse”. Does it mean that traditional models tend to fail to capture “one-to-many" polysemy representations? If so, what supports this claim?"**
>
> **Response:**
> Thank you for the question. Yes—our statement is closely related to the difficulty of point (deterministic) embeddings in capturing the “one-to-many” nature of medical semantics (polysemy, incomplete descriptions, and uncertainty), but we want to clarify what we mean by “embedding space collapse” and what supports it in our paper.
>
> In our setting, “traditional deterministic embeddings” (e.g., CLIP-style cosine alignment with a single vector per image/text) force each input to be represented as one fixed point. When the supervision contains inherent ambiguity—e.g., the same imaging pattern can be described by multiple valid terms (“nodule” vs. “mass”), and reports can omit normal organs (“unmentioned ≠ non-existent”)—the learning objective can push many heterogeneous samples toward overly similar representations to satisfy multiple inconsistent pairings. This phenomenon is commonly discussed in probabilistic embedding literature as an over-concentration / reduced diversity of embeddings when the model cannot express uncertainty or multi-modality with a single point.
>
> Our paper supports this claim in two ways:
>
> Motivation grounded in the data characteristics. In the Introduction (Sec. 1), we explicitly describe that medical image-text pairs exhibit “one-to-many” polysemy and semantic incompleteness. A single-point mapping has limited capacity to represent a set of plausible meanings, whereas a distribution can encode both a central tendency (mean) and a confidence/ambiguity level (variance).
>
> Empirical evidence via controlled comparison and ablations. We retrain the deterministic CT-CLIP baseline from scratch under the same hyperparameters (Sec. 4.2) to isolate architectural effects. Replacing point embeddings with our probabilistic contrastive objective (PPCL Global) yields consistent gains in zero-shot detection (Table 1 vs. Table 3: Accuracy/F1/AUROC from 0.643/0.680/0.679 to 0.656/0.691/0.702). This indicates that modeling representations as distributions (with variance) improves robustness under ambiguous supervision. Additionally, the full framework further improves performance when we add local alignment and hierarchical inclusion constraints (Tables 3–4), suggesting that uncertainty-aware representations plus structured constraints better handle fine-grained, multi-granularity semantics than deterministic global alignment.
>
> We will revise the wording in the Introduction to be more precise: rather than claiming that collapse is always inevitable, we mean that deterministic point embeddings are prone to over-concentration and insufficient expressiveness under one-to-many / uncertain supervision, and our probabilistic formulation is designed to mitigate this by explicitly modeling uncertainty with variance (Sec. 3.3) and enforcing hierarchical consistency.
>
> **For Question 2: "What size of patch do you use for the 3D Vision Transformer? If different patch size were tried, have you noticed any difference in performance?"**
>
> **Response:**
> Thank you for the question. We use a 3D patch size of 16 × 16 × 8. This choice follows the commonly used 16 × 16 patching in 2D ViTs and adapts the third dimension to CT characteristics: the inter-slice spacing / slice thickness is typically larger than the in-plane pixel spacing, so we use a coarser patching along the depth axis.
>
> We also considered using a smaller depth patch size (i.e., reducing the third dimension below 8) to increase through-plane resolution. However, this would substantially increase the number of tokens and lead to a steep rise in GPU memory and compute (token count grows inversely with patch volume, and attention cost grows super-linearly with the number of tokens). On the other hand, using a much larger depth patch would mismatch the slice thickness scale and may overly smooth through-plane details. Overall, 16 × 16 × 8 provides a practical trade-off between anatomical fidelity and computational feasibility in our setting.

---

> > ### Author Response · Authors · 2026-01-25
> > **Rebuttal to Reviewer nPCa PART Ⅱ**
> >
> > **For Question 3: "Why is a 3D Vision Transformer not included as a supervised baseline?"**
> >
> > **Response:**
> > Thanks for your suggestion. We have added the result of 3D Vision Transformer in our paper.
> >
> > **For Question 4: "CT-GLIP is mentioned in the related work section. Why isn’t it included in the experimental results?"**
> >
> > **Response:**
> > Thanks for your suggestion. The CT-GLIP model and its training data are not open-sourced, so we did not include it in our comparative experiments.
> >
> > **For Question 5: "I am not sure to clearly understand why/how the diagonal covariance matrix introduced in 3.3 represents the uncertainty. Please, would the authors give me more explanations on it?"**
> >
> > **Response:**
> > In Sec. 3.3 we represent each image/text embedding as a diagonal Gaussian distribution $p(z|x)=\mathcal{N}(\mu(x),\Sigma(x))$ with $\Sigma(x)=\mathrm{diag}(\sigma^2(x))$. In this formulation, the mean $\mu$ encodes the central (most likely) semantics, while the covariance controls how “spread out” the representation is in the embedding space. A larger variance $\sigma_i^2$ along a dimension means the model is less confident about the corresponding latent factor, so samples from $p(z|x)$ will vary more, yielding a broader region of plausible embeddings—this is precisely what we refer to as uncertainty.
> > This uncertainty has a direct effect on both similarity computation and learning. In our probabilistic similarity/logit (Eq. (6)), the term $-\tfrac{1}{2}\mathrm{Tr}(\Sigma_v+\Sigma_t)$ penalizes overly uncertain pairs: when an input is ambiguous (e.g., polysemous wording, incomplete report, blurry lesion boundary), the model can increase $\Sigma$ to avoid committing to a single sharp point, but this comes at a cost in the alignment score. Thus, the variance is not a free parameter; it is learned to balance data fit vs. confidence. Additionally, our VIB regularizer (Eq. (8)) prevents degenerate solutions (e.g., driving variances to extreme values) and encourages well-calibrated uncertainty.
> > We use a diagonal covariance for a practical trade-off: it captures per-dimension (anisotropic) uncertainty while remaining stable and efficient in high-dimensional embeddings, avoiding the heavy computation and potential instability of full covariance estimation.

---

> ### Author Response · Authors · 2026-01-25
> **Rebuttal to Reviewer nPCa PART Ⅲ**
>
> **For Comment 1: "While authors address the challenges of global-local alignment, the zero-shot multi-abnormality detection evaluation would benefit from including 3D-ViT, the HiPro-CT visual encoder. In link with learning robust representations both for short- and long-range dependencies, integrating CT-Scroll (MIDL, 2025), a recently introduced 2.5D backbone that leverages local and global attentions would eventually strengthen the quantitative comparison."**
>
> **Response:**
> Thank you for the valuable suggestion. We have updated the zero-shot multi-abnormality detection evaluation by adding an additional baseline experiment as requested, and we include the corresponding quantitative results and discussion in the revised manuscript.
>
> **For Comment 2: "As the loss function is composed of four terms, a leave-one-out ablation study on auxiliary terms would complement the ablation study on the contrastive loss."**
>
> **Response:**
> We agree that, since our total objective includes four terms, a leave-one-out ablation on the auxiliary losses can better clarify their individual contributions. We have conducted and added this leave-one-out ablation study in the revised version.
>
> **For Comment 3: "Qualitative results examples of correct prediction for zero-shot abnormality detection, eventually complemented by overlayed activation (or attention, similarity...) maps would help to see on which regions do the model focus."**
>
> **Response:**
> Thank you for this suggestion. We have added qualitative examples of correct zero-shot abnormality predictions and complemented them with Grad-CAM visualizations to illustrate the model’s localization behavior. These results are provided in Appendix C.
>
> **For Comment 4: "No qualitative results are included, limiting evaluation depth to quantitative tables."**
>
> **Response:**
> We agree that per-abnormality comparisons are important for understanding whether gains are consistent across findings. We have included per-abnormality performance with a direct comparison to CT-CLIP in Appendix A.
>
> **For Comment 5: "As the experimental results focuses on a single dataset institution, performing a cross-dataset evaluation for zero-shot abnormality detection would strengthen claims, for example on RadChestCT (TMI 2021)."**
>
> **Response:**
> We agree that cross-dataset evaluation would strengthen the generalization claims. However, we are unfortunately unable to include RadChestCT (TMI 2021) in this revision because our data access request was rejected due to recent changes in U.S. policies. We have noted this limitation and will pursue additional cross-dataset evaluations in future work when data access becomes possible.

---

### Author Rebuttal · Authors · 2026-01-25

**Rebuttal:**

We thank the reviewers for their insightful comments and have addressed the key concerns as follows. Please refer to the revised manuscript in the attached PDF (Supporting Material).

1. In response to Reviewer nPCa's Question 1, 2, 4 and 5, we've posted the response on the comment board.

2. In response to Reviewer nPCa's Question 3 and Comment 1, we have conducted supplementary experiments and incorporated the relevant results into the revised manuscript.

3. In response to Reviewer nPCa's Comment 2, we have conducted supplementary experiments and incorporated the relevant results in the Table 3 and Table 4 in the revised manuscript.

4. In response to Reviewer nPCa's Comment 3, we have conducted supplementary experiments and incorporated the relevant results in the Appendix C.

5. In response to Reviewer nPCa's Comment 4, we have conducted supplementary experiments and incorporated the relevant results in the Appendix A.

6. In response to Reviewer eX25's Question 1, our original manuscript reflects the relevant results, and we have further emphasized them in the revised version.

7. In response to Reviewer eX25's Question 2, we have conducted supplementary experiments and included the results in Appendix B.

8. In response to Reviewer DqeW's Question, we have provided a detailed explanation in the conment board and included additional results in the revised manuscript as suggested by the reviewer.

9. We have responded to all reviewers' comments and questions via the online comment board.

**Supporting Material:**

/attachment/43676d0e227033130b92c8b9d8e0228b68306183.pdf

---

> ### Author Response · Authors · 2026-01-29
>
> We updated all results with 95%CI (through 1000 bootstraps).

---

### Meta-Review · Area_Chair_Jn6u · 2026-02-03

**Recommendation:** Accept (Poster)
**Confidence:** 4

**Metareview:**

Thank you to the authors and reviewers for engaging in discussion. The final reviewer scores are strong accept, weak accept, and weak reject. Reviewer nPCa's concerns supporting their weak reject are valid, but I don't believe that they disqualify the paper from MIDL, although I won't recommend it for an oral.

---

### Decision · Program_Chairs · 2026-02-13

Accept (Poster)